# An Efficient Recognition Method for Orbital Angular Momentum via Adaptive Deep ELM

**DOI:** 10.3390/s23218737

**Published:** 2023-10-26

**Authors:** Haiyang Yu, Chunyi Chen, Xiaojuan Hu, Huamin Yang

**Affiliations:** School of Computer Science and Technology, Changchun University of Science and Technology, Changchun 130022, China; chenchunyi@cust.edu.cn (C.C.); huxj@cust.edu.cn (X.H.); yhm@cust.edu.cn (H.Y.)

**Keywords:** optical communication, orbital angular momentum, atmosphere turbulence, multilayer ELM

## Abstract

For orbital angular momentum (OAM) recognition in atmosphere turbulence, how to design a self-adapted model is a challenging problem. To address this issue, an efficient deep learning framework that uses a derived extreme learning machine (ELM) has been put forward. Different from typical neural network methods, the provided analytical machine learning model can match the different OAM modes automatically. In the model selection phase, a multilayer ELM is adopted to quantify the laser spot characteristics. In the parameter optimization phase, a fast iterative shrinkage-thresholding algorithm makes the model present the analytic expression. After the feature extraction of the received intensity distributions, the proposed method develops a relationship between laser spot and OAM mode, thus building the steady neural network architecture for the new received vortex beam. The whole recognition process avoids the trial and error caused by user intervention, which makes the model suitable for a time-varying atmospheric environment. Numerical simulations are conducted on different experimental datasets. The results demonstrate that the proposed method has a better capacity for OAM recognition.

## 1. Introduction

Optical carrier technology has received great attention for high-rate information transmission [1]. The orbital angular momentum (OAM) provides abundant carrier information, which is widely used in Free Space Optical (FSO) communication systems [2]. However, as an important factor in FSO, atmosphere turbulence causes the distortion of optical signals. It will lead to the propagation phenomena such as laser spot scintillation, laser beam drift, and arrival angle fluctuation [3]. These negative effects increase the difficulty of optical signal processing. If the received laser spot information cannot be extracted effectively, the transmission capacity and spectral efficiency will be limited [4]. Therefore, how to recognize the OAM modes in different atmosphere turbulence environment has been regarded as a challenging problem.

Because the vortex beam with different superposition states presents various physical phenomena, the OAM mode recognition in turbulences can extend the channel capacity of communication systems. Taking advantage of the data-driven strategy, many researchers devote themselves to developing the OAM mode detection schemes. For example, Krenn et al. [5] firstly use the Artificial Neural Network (ANN) as an OAM mode classification. Then, Krenn’s team conducts four OAM recognition experiments in a maritime atmospheric channel and confirms that machine learning can support the information transmission over hundreds of kilometers [6]. Sun et al. use support vector machine (SVM) to classify the OAM of a Laguerre–Gaussian beam [7]. In order to express the subtle features, deep learning is usually used as a modeling tool for unstructured data [8]. Jing et al. propose a fractional OAM mode recognition method with a neural network frame, and extend the original optical features [9]. Meanwhile, Fu et al. propose a hybrid interference–convolutional neural network scheme [10]. They use a deep learning framework to analyze atmosphere turbulence and improve recognition performance under different turbulence levels. Driven by miscellaneous applications, Zhao et al. propose an OAM detection strategy based on a diffractive multilayered neural network [11]. Furthermore, Zhou et al. retrain a convolutional neural network (CNN) to explore hidden OAM information in big data environments [12]. A spatial diversity turbulence mitigation scheme is designed to maintain the performance of an OAM-multiplexed communication link [13]. It is obvious that the machine learning approach can achieve the OAM mode classification effectively. However, the complex iteration process of parameters makes it difficult for the neural network to obtain an analytical solution. It causes trouble to build the appropriate model for different dynamic turbulence scenes. 

Recently, as a typical feed-forward neural network, extreme learning machine (ELM) has significant advantages including the theoretical analytical solution and fast convergence speed, thus improving the feasibility of image processing [14]. In the training stage, ELM calculates the output weight by Moore–Penrose (M-P) generalized inverse, which avoids the complicated parameter ergodic process [15]. In a similar approach to [15], Huang et al. [16] use a penalty function to balance model accuracy and complexity. This penalty regression ELM overcomes the defect that the least square method is not suitable for nonlinear fitting. Tavakoli et al. attempt to develop a novel fixation prediction framework based on inter-image similarities [17]. The ELM model estimates the saliency of the given image and proves the effect of image classification. At an atmospheric turbulent environment, the recognition model of OAM mode will be appropriate for the changing scenario. To acquire the expected generalization performance, the error minimized ELM (EM-ELM) moves toward the predictive target with varying hidden node [18]. Then, D-ELM has a huge advantage on data fitting when the target function is not explicit, and the compact input set reduces the model size [19]. A recent multilayer perceptron method [20] shows a significant performance improvement for complex feature mapping. In a word, think of the intricate relationship between laser spot and OAM mode, where the updated mechanism of recognition model is a vital condition to guarantee the global optimum.

Different from the above previous work, in this paper, we present an efficient OAM recognition method with adaptive deep ELM (AD-ELM). Expanding the original ELM, the proposed method not only describes the subtle feature, but also fully considers the relationship between OAM and laser spot. Also, in order to accelerate the calculation processing, we bring in the fast iterative shrinkage-thresholding algorithm (FISTA) [21] to obtain convergence results. 

This paper contains several contributions as follows:(1)Different from the previous job, this paper not only considers the randomness of parameters in the neural network as a whole [22], but also obtains the minimum norm constraint of the global solution.(2)Then, the updated model will be self-driven, that means OAM mode recognition will obtain the analytical expression. The whole learning process avoided manual parameter tuning.(3)It has significant application value for OAM mode recognition according to the atmospheric turbulent environment.

The whole thesis has four main parts. Part 1 is the introduction. Part 2 involves atmosphere turbulence and ELM. Part 3 interprets the proposed AD-ELM method. The last part is the results.

## 2. Preliminaries

The proposed method takes root in the atmosphere turbulence theory and ELM model. The atmosphere turbulence theory is the key condition to guarantee the performance and efficiency of wireless optical communication systems. The ELM has a complete theoretical system and easy hardware implementation, and it becomes feasible for unstructured data processing and image recognition [23]. In this section, the two parts are outlined.

### 2.1. Atmosphere Turbulence Theory

The atmospheric refractive index is one of the most efficient means to represent atmospheric conditions [24]. The structure constant of the refractive index $C_{n}^{2}$ helps us to describe atmosphere turbulence. Based on the Kolmogorov theory [25], for the isotropic turbulence, the refraction index $D_{n}\left(r\right)$ can be described by the $C_{n}^{2}$, which is shown as follows:(1)$$D_{n}\left(r\right)=\langle {\left[n\left(r+r_{1}\right)-n\left(r_{1}\right)\right]}^{2}\rangle =C_{n}^{2}r^{2/3},\;l_{0}<r<L_{0},$$
where $r_{1}$ denotes the initial position, n denotes the refractive index, $l_{0}$ is the internal scale, and $L_{0}$ is the external scale. The symbol 〈⋅〉 represents the average value. For example, the Hufnagel–Valley model expresses the daytime atmosphere turbulence. In this model, the atmospheric refractive index and wind speed are adjustable. They fit various site conditions to match specific values for the coherence length and the isoplanatic angle [26]. Assuming that the laser is traveling forward along the z axis, the atmospheric coherent parameter $r_{0}$ can be expressed as [27]
(2)$$r_{0}={\left[0.423k^{2}\int_{0}^{\Delta_{z}}C_{n}^{2}\left(z\right)dz\right]}^{-5/3},$$
where Δz is the thickness of each thin layer. We can find that the parameter $r_{0}$ describes the phase disturbance intensity. The smaller $r_{0}$ is, the stronger the disturbance becomes.

### 2.2. Brief of ELM Model

According to the ELM construction, the weights between the input layer and hidden layer are determined randomly, and the output vector is calculated analytically [28]. Suppose that the sample data ${\left\{\left(x_{i},y_{i}\right)\right\}}_{i=1}^{N}$, the mapping relationship can be expressed as
(3)$$y_{i}=\sum_{p=1}^{L}\beta_{p}\varphi \left(a_{p}\cdot x_{i}+b_{p}\right),\;i=1,\ldots ,N,$$
where L is the number of hidden nodes, $a_{p}$ is the input weight vector, and $b_{p}$ is the offset. $\beta_{p}$ is the learning parameter connecting the pth hidden node to the output node. Normally, in a more concise expression, the above expression can be written in a matrix form as follows:(4)Y=Hβ,
where $H={\left[\begin{array}{ccc} \varphi \left(a_{1}\cdot x_{1}+b_{1}\right) & \ldots & \varphi \left(a_{L}\cdot x_{1}+b_{L}\right)\\ \vdots & \ldots & \vdots \\ \varphi \left(a_{1}\cdot x_{N}+b_{1}\right) & \ldots & \varphi \left(a_{L}\cdot x_{N}+b_{L}\right) \end{array}\right]}_{N\times L}$, $\beta ={\left[\beta_{1},\beta_{2},\ldots ,\beta_{L}\right]}^{Τ}$ and $Y={\left[y_{1},y_{2},\ldots ,y_{N}\right]}^{Τ}$. Since there is no iteration operation at solution procedure, the ELM is faster than the traditional gradient descent method.

## 3. Preliminaries

In this section, we introduce the proposed AD-ELM in detail. This method includes two phases. One of them is the model selection, and the other one is the parameter estimation. With the basic ELM, the deep learning is selected to establish a suitable OAM recognition model. We implemented numerical simulation and received light spots through a phase screen. The schematic diagram of the method is illustrated in Figure 1.

### 3.1. Model Selection Phase

Model selection is necessary to obtain the OAM analytic expression by means of the data-driven theory. We have simulated the Laguerre–Gaussian (LG) beams, which propagated in the random atmosphere turbulence channel. The complex amplitude expression is shown as follows:(5)$$U_{l}\left(r,\varphi ,z\right)=R_{l}\left(r,z\right)\exp\left(il\varphi \right),$$
where i is the imaginary unit, and l is the topological charge. Otherwise, l is known as spatial modes. The number of l is the OAM mode label. exp(ilφ) is the phase factor. The radial basis function $R_{l}\left(r,z\right)$ is then given by
(6)$$\begin{array}{l} R_{l}\left(r,z\right)=\sqrt{\frac{2p!}{\pi \left(p+\left|l\right|\right)!}}\frac{1}{\omega \left(z\right)}{\left[\frac{r\sqrt{2}}{\omega \left(z\right)}\right]}^{\left|l\right|}L_{p}^{l}\left[\frac{2r^{2}}{\omega^{2}\left(z\right)}\right]\exp\left[\frac{-r^{2}}{\omega^{2}\left(z\right)}\right]\\ \;\;\;\;\times \exp\left[\frac{-ikr^{2}z}{2\left(z^{2}+z_{R}^{2}\right)}\right]\exp\left[i\left(2p+\left|l\right|+1\right)\arctan\left(\frac{z}{z_{R}}\right)\right] \end{array},$$
where p is the radial index, $\omega \left(z\right)=\omega_{0}\sqrt{1+\left(z^{2}/z_{R}^{2}\right)}$, $\omega_{0}$ is the waist radius, $z_{R}=\pi \omega_{0}^{2}/\lambda$ is the Rayleigh distance, λ is the wavelength, and $L_{p}^{l}\left(\cdot \right)$ is the Laguerre polynomial. Considering the radial index p=0 and topological charge number $\pm l_{0}$, the intensity of light field can be further expressed according to the Helmholtz equation
(7)$$\begin{array}{l} {\left|U_{\pm l_{0}}\left(r,z\right)\right|}^{2}={\left|U_{l_{0}}\left(r,\varphi ,z\right)+U_{-l_{0}}\left(r,\varphi ,z\right)\right|}^{2}\\ \;\;\;\;\;\;={\left|R_{l_{0}}\left(r,z\right)\exp\left(il_{0}\varphi \right)+R_{-l_{0}}\left(r,z\right)\exp\left(-il_{0}\varphi \right)\right|}^{2}\\ \;\;\;\;\;\;=2{\left|R_{\left|l_{0}\right|}\left(r,z\right)\right|}^{2}\left(1+\cos\left(2l_{0}\varphi \right)\right) \end{array},$$
where the ${\left|R_{\left|l_{0}\right|}\left(r,z\right)\right|}^{2}$ can be expressed as
(8)$${\left|R_{\left|l_{0}\right|}\left(r,z\right)\right|}^{2}=\frac{2}{\pi \left|l_{0}\right|!}\frac{1}{\omega^{2}\left(z\right)}\exp\left[-2\left(\frac{-r^{2}}{\omega^{2}\left(z\right)}\right)\right]{\left[\frac{\sqrt{2}r}{\omega \left(z\right)}\right]}^{2\left|l_{0}\right|}.$$

Usually, the proposed method generates the random phase screen based on fast Fourier transform (FFT) and inverse fast Fourier transform (IFFT) [29], which is shown as follows:(9)$$\begin{array}{c} U\left(r_{1}\right)=F^{-1}\left[F\left[U\left(r_{0}\right)\right]\cdot \exp\left[G\right]\right]\exp\left[i\phi \left(r_{0}\right)\right]\\ U\left(r_{2}\right)=F^{-1}\left[F\left[U\left(r_{1}\right)\right]\cdot \exp\left[G\right]\right]\exp\left[i\phi \left(r_{1}\right)\right]\\ \vdots \\ U\left(r_{m}\right)=F^{-1}\left[F\left[U\left(r_{m-1}\right)\right]\cdot \exp\left[G\right]\right]\exp\left[i\phi \left(r_{m-1}\right)\right] \end{array},$$
where $U\left(r_{0}\right)$ is the intensity of the light field at the source, $U\left(r_{1}\right)$ is the intensity of the light field when the beam has passed through a phase screen, and $U\left(r_{m}\right)$ represents the light field intensity of the received plane. F and $F^{-1}$ express the Fourier transform and the inverse Fourier transform, $G=i\Delta z\left(2g^{2}-g_{r}^{2}\right)/\left(2g\right)$ is transformation operator, Δz is the distance of phase screen at each layer, $g_{r}^{2}$ is the number of spatial waves, and $r_{0}=\left(x_{0},y_{0}\right),r_{1}=\left(x_{1},y_{1}\right),\cdots ,r_{m}=\left(x_{m},y_{m}\right)$ denote coordinates. Here, the grid size of the laser spot is 256×256. $C_{n}^{2}=r_{0}{}^{\left(-5/3\right)}/0.423k^{2}$ is equal to the structure constant of the refractive index. k=2π/λ is the number of laser beams. λ is the wavelength as above. After N times Monte Carlo simulation, the sample set of laser spots is $P_{M,N}=\left\{U_{i}\left(r_{m}\right)|i=1,2,\cdots N,M=m\times m\right\}$. Then, the laser spot $P_{M,N}$ can be converted to feature vectors $P_{M,N}=\left[p_{M,}{}_{1},p_{M,}{}_{2},\cdots ,p_{M,}{}_{N}\right]$. The OAM mode label is $Y_{N}={\left[y_{1},y_{2},\ldots ,y_{N}\right]}^{Τ}$. The training set can be described as $\left[P_{M,N},Y_{N}\right]={\left\{\left(p_{M,}{}_{i},y_{i}\right)\right\}}_{i=1}^{N}$.

After receiving the sample set of laser spots, the relationship between the feature vectors and OAM mode label can be calculated by multilayer ELM. The expression can be defined analytically:(10)$$Y_{N}=H\beta =H_{1}\beta_{1}\beta_{2}\cdots \beta_{J},$$
where $\beta =\left(\beta_{1},\beta_{2},\cdots \beta_{j},\cdots \beta_{J}\right)$ is the output weight, and $H=\left(H_{1},H_{2},\cdots H_{j},\cdots H_{J}\right)$ is the hidden output. $H_{1}$ shows the polynomial structure constituted by weight $a_{1}={\left[a_{1,1},a_{1,2},\ldots ,a_{1,L_{1}}\right]}^{T}$ and offset $b_{1}={\left[b_{1,1},b_{1,2},\ldots ,b_{1,L_{1}}\right]}^{T}$, which can be expressed as
(11)$$H_{1}=h\left(P_{M,N}\right)={\left[\begin{array}{ccc} \varphi \left(a_{1,1}\cdot p_{M,}{}_{1}+b_{1,1}\right) & \cdots & \varphi \left(a_{1,L_{1}}\cdot p_{M,}{}_{1}+b_{1,L_{1}}\right)\\ \vdots & \cdots & \vdots \\ \varphi \left(a_{1,1}\cdot p_{M,}{}_{N}+b_{1,1}\right) & \cdots & \varphi \left(a_{1,L_{1}}\cdot p_{M,}{}_{N}+b_{1,L_{1}}\right) \end{array}\right]}_{N\times L_{1}}.$$

The model contains multiple hidden layers. Meanwhile, each of them has independent weights. Multilayer mapping extracts feature sets automatically. Thus, the output of each hidden layer is determined by the previous layer value, which can be expressed as
(12)$$H_{j}=H_{j-1}\cdot \beta_{j-1},$$
where $H_{j}$ and $H_{j-1}$ represent the output matrix for the *j*th layer and (j−1)th layer.

The current feature becomes the input element of the next layer. When the previous layer confirms mapping relationship, the current vector is determined as well. Then, the weight expression of the output layer will be obtained. In order to improve the generalization ability, we added norm constraints to each hidden layer. The optimization object function is transformed into the form minimization between the approximation error and regularization, which can be expressed as
(13)$$\varepsilon =F\left(\beta \right)=\arg\;\underset{\beta }{\min}\left\{C{\left\|H\beta -Y_{N}\right\|}^{2}+{\left\|\beta \right\|}_{1}\right\},$$
where C is the regularization parameter. It can balance the approximation error and model adaptability.

### 3.2. Parameter Estimation Phase

In order to achieve a parameter estimation of the output layer, the Fast Iterative Shrinkage-Thresholding Algorithm (FISTA) [21] is selected to accelerate the calculation. Taking the whole hidden layers into consideration, the Jth output vectors will be calculated, and the approximation error part can be abstracted to g(β)
(14)$$g\left(\beta \right)={\left\|H\beta -Y_{N}\right\|}^{2},$$
where $\beta =\left(\beta_{1},\beta_{2},\cdots \beta_{j},\cdots \beta_{J}\right)$ is output weight. The auxiliary parameter s is the output weight of the previous layer, which means $s_{j}=\beta_{j-1}\in R^{n}$. After determined the interval of each iteration, the relationship between $\beta_{j}$ and $s_{j}$ becomes
(15)$$\beta_{j}=PL\left(s_{j}\right),$$
where PL(⋅) specific expression is described according to the minimum lower bound of the derivative. When the gradient of approximation function satisfies the Lipschitz condition, there exists a constant $L_{P}$ satisfying the following formula:(16)$${\left\|\nabla g\left(\beta \right)-\nabla g\left(s\right)\right\|}_{2}^{2}\leq L_{P}{\left\|\beta -s\right\|}_{2}^{2}\;,\forall \beta ,s\in R^{n},$$
where ∇ represents the differential operator. Using Taylor formula expansion and gradient descent method, we obtain auxiliary sequence $s_{j+1}=s_{j}-\frac{1}{L_{P}}\nabla f\left(s_{j}\right)$. It helps calculate the optimal solution of F(β). The quadratic minimization form is shown as follows:(17)$$PL\left(s\right)=\arg\;\underset{\beta }{\min}\left\{g\left(s\right)+\langle \beta -s,\nabla g\left(s\right)\rangle +\frac{L_{P}}{2}{\left\|\beta -s\right\|}_{2}^{2}+{\left\|\beta \right\|}_{1},\forall \beta ,s\in R^{n}\right\}.$$

After ignoring the irrelevant constant term, the simplified optimization equation can be obtained as
(18)$$PL\left(s\right)=\arg\;\underset{\beta }{\min}\left\{\frac{L_{P}}{2}{\left\|\beta -\left(s-\frac{1}{L_{P}}\cdot \nabla f\left(s\right)\right)\right\|}^{2}+{\left\|\beta \right\|}_{1}\right\}.$$

During the iteration of each layer, the appropriate step size $\theta_{j}$ of each layer is calculated from the gradient, with ∇f being a smooth convex function. $\theta_{1}=1$ is the initial point. For each output layer of the neural network, the above equation can be converted to the following formula:(19)$$PL\left(s_{j}\right)=\arg\;\underset{\beta }{\min}\left\{\frac{1}{2\theta_{j}}{\left\|\beta -\left(s_{j}-\theta_{j}\cdot \nabla f\left(s_{j}\right)\right)\right\|}^{2}+{\left\|\beta \right\|}_{l_{1}}\right\}.$$

In order to maintain the convergence, the appropriate step size $\theta_{j}$ can be calculated recursively from the parameter values of the previous layer:(20)$$\theta_{j}=\frac{1+\sqrt{1+4\theta_{j-1}^{2}}}{2}.$$

The update expression of the auxiliary parameter $s_{j}$ can be expressed as
(21)$$\beta_{j}=PL\left(s_{j+1}\right)=PL\left(\beta_{j}+\left(\frac{\theta_{j}-1}{\theta_{j+1}}\right)\left(\beta_{j}-\beta_{j-1}\right)\right)$$

After the iterative steps, the parameters of multilayer ELM were finally determined. Considering the characteristics of the learning procedure, the whole AD-ELM algorithm is presented in Algorithm 1.

**Algorithm 1:** Learning Procedure of the AD-ELM
1. Generate Laguerre Gaussian beams2. Calculate the Phase Screen at objective position3. Building the mapping relationship between laser spots and OAM mode4. Establish multilayer ELM network structure5. Solve the output weight of each layer6. Return the analytic solution of the output weight7. Identify OAM in the new simple set

## 4. Simulation Results and Discussion

In this section, the simulation results have been described in detail. The dataset consists of four simulation conditions. Then, the proposed AD-ELM is compared with four other classical OAM recognition methodologies.

### 4.1. Dataset Generation

To validate the effectiveness of the proposed method, the continuous phase screen is chosen to describe the atmospheric condition. We simulate the 2000m transmission distance. The visualized laser spot diagram is displayed in Figure 2.

Figure 2a–d shows the original laser spot and received laser spot for single OAM and multiplexed OAM. Davis proposed a method to divide the strength of turbulence according to $C_{n}^{2}$ [30]. When the $C_{n}^{2}<6.4\times 10^{-17}\;m^{-2/3}$, the atmospheric turbulence is a weak turbulence. As $C_{n}^{2}$ increases, the atmospheric turbulence gradually becomes stronger. In order to evaluate the effectiveness of the proposed method, we generate different datasets in which $C_{n}^{2}$ are $1\times 10^{-14}\;m^{-2/3}$ and $1\times 10^{-15}\;m^{-2/3}$, respectively. After image processing, the received laser spot images retain the key characteristic information, and the spot size of each attribute is 28×28. We simulated 2500 OAM modal samples. Here, the 2000 training samples and 500 test samples are randomly collected as the input of the model. For the single topological charges, the ten OAM modes that we use in our simulation process are given by $\left|l_{0}\right|=1,2,\ldots 10$. And for the multiplexed topological charges, the eight OAM modes that we use in our simulation are given by $\left|l_{0}\right|=1,2,\ldots 8$. The intensity images of the received LG beam in $1\times 10^{-14}\;m^{-2/3}$ are shown in Figure 3.

Then, all the features must be normalized into the range [−1,1]. The detailed formula is shown as follows:(22)$$x^{\prime }=2\times \frac{x-x_{\min}}{x_{\max}-x_{\min}}-1,$$
where x is an original numeric value, $x_{\min}$ is the minimum of the current character data, and $x_{\max}$ is the maximum of the current character data. Once the training process is completed, the model will perform a recognition operation at the test period.

### 4.2. Evaluation Criteria

The generalization performance for various methods is analyzed with the Root Mean Square Error (RMSE) and G-mean. The RMSE can be computed from the following criteria.
(23)$$\mathrm{RMSE}=\sqrt{\frac{1}{N_{t}}\sum_{i=1}^{N_{t}}{\left(y_{i}-{\hat{y}}_{i}\right)}^{2}}.$$
where $y_{i}$ is the *i*th actual value, $i=1,\ldots ,N_{t}$. ${\hat{y}}_{i}$ is the calculated value by the model, $N_{t}$ is the number of testing records. The G-mean is calculated by the confusion matrix in Table 1.

For the whole dataset with $M_{t}$ labels, they are shown as the following forms:(24)$$G-mean={\left(\prod_{i=1}^{M_{t}}\frac{TP_{i}}{TP_{i}+FN_{i}}\right)}^{\frac{1}{M_{t}}}$$

The higher G-mean, the better recognition performance for multi-classification data.

### 4.3. Simulation Results

The whole forecasting process is executed by 20 Monte Carlo trials. The BP-ANN, SVM, kNN and DNN are chosen to be the contrastive methods. We firstly analyzed the recognition result for different OAM modes. Then, the visualized classification distribution result of the proposed method is displayed in Figure 4.

We found the various categories in Figure 4. The different color spots with various color circles show the different datasets. The pink circle, gray circle, blue circle, dark blue circle, light blue circle, dark green circle, green circle, light green circle, purple circle and orange circle indicate the correct identification from the first mode to the tenth mode. Triangles of the corresponding color indicate false identification. We first calculated the principal components, and then sorted them from the largest to the smallest. After calculating the eigenvectors for each sample separately, the coordinates in two directions represent the two largest principal component scores, respectively. The proposed method distinguishes the different species. Further discussion of the results, the RMSE and G-mean of the proposed method and the comparative methods are shown as the following figure. Figure 5 shows the average RMSE for the two turbulence scenarios as various color lines. We use blue bars to represent the proposed method, and the red ones, the yellow ones, the green ones, and the pink ones represent BPP-ANN, SVM, kNN, and DNN, respectively.

In Figure 5, we can find that the proposed method has the minimum RMSE, which achieves 0.1732 and 0.0894 for $C_{n}^{2}=1\times 10^{-14}\;m^{-2/3}$ and $C_{n}^{2}=1\times 10^{-15}\;m^{-2/3}$ environment, respectively. It is proved that the proposed AD-ELM has better adaptive capacity. This advantage will describe the model analytically based on the global optimal parameters. To reflect the predicted multiple OAM recognition performance, we calculated the G-mean of the five methods, which is shown in Figure 6.

From Figure 6a,b, it is clear that the G-mean of the proposed method is always at the top position, which achieves 0.9585 and 0.9915. Especially in Figure 6a, the G-mean of the proposed method is close to 1, which is far higher than the constructive methods. The kNN shows good performance in a weak turbulence environment. If the turbulence becomes strong, the performance will degrade dramatically. It should be noted that due to the controlling weight parameters along with the turbulence variation, the proposed method predicts a high-quality annular shape effectively. What is more, the performances of recognizing multiplexed OAM mode sets are further discussed. Figure 7 shows the visualized classification distribution of the proposed method for eight OAM modes.

Following the simulation setup shown in Figure 7, we can find that the proposed method distinguishes the different categories effectively. The different color spots with various color circles show the different datasets. The pink circle, gray circle, blue circle, dark blue circle, light blue circle, dark green circle, green circle, light green circle, purple circle and orange circle indicate the correct identification from the first mode to the tenth mode. Triangles of the corresponding color indicate false identification. The sample presents dispersed distribution in strong turbulent environment. When the turbulence becomes weak, the distinguishable spatial distribution appears. Then, Figure 8 shows the RMSE of the five methods for multiplexed topological charges dataset.

Analysis of RMSE sees that the proposed method has smaller generalization errors in the whole testing process, which are 0.3847 and 0.0632. If the $C_{n}^{2}$ becomes higher, the changeable weight will match the nonlinear feature effectively. Because of the quantitative analytical expression between the adjacent layers, the parameters will be self-adapted and the performance fluctuation is not violent. Then, we measured the G-mean for multiplexed topological charges dataset, which is shown in Figure 9.

From Figure 9, we can find that the G-mean of proposed method is 0.7849 and 0.9979. It considered different turbulent environments. As seen in Figure 9b, the contrastive methods show the good value in the weak turbulence environment. When the turbulence turns strong, the proposed AD-ELM also reaches the optimal model to match the changing environment. At the same time, a better analytical expression is given by the AD-ELM, which represents a higher model capacity without manual trial and error operation, which have significant application values for long-term online recognition. The optimal parameters help our method be more suitable for the regular OAM mode sets.

From the simulation result, the proposed method receives better generalization performance. It is because holding the optimal parameters acquires the optimal analytic solution. So, the OAM recognition will be supported by the adaptive model.

## 5. Conclusions

In this paper, an efficient AD-ELM is proposed for orbital angular momentum recognition. Different from a traditional neural network, the proposed method quantified the laser spot characteristics. We brought a fast iterative shrinkage-thresholding into the iterative solution procedure. We receive integrated weight automatically. So, the whole process of OAM recognition requires no manual trial and error. Compared with several contrasting methods, the simulation result shows the proposed method has better recognition results. It indicated the method matches the new facula sample well. The OAM with high-dimensional feature space will be researched in the future.

## Figures and Tables

**Figure 1 sensors-23-08737-f001:**
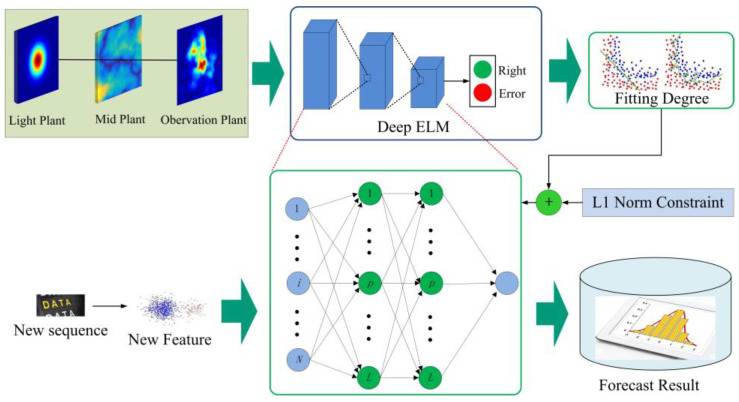
Schematic diagram of the proposed method.

**Figure 2 sensors-23-08737-f002:**
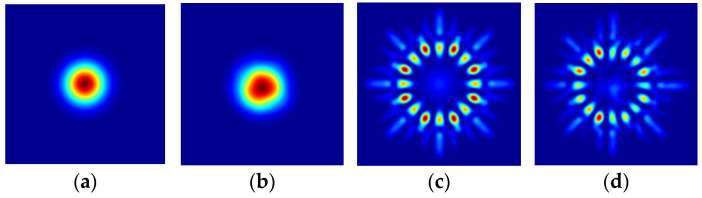
Laser spot diagram. (**a**) Single original laser spot; (**b**) single received laser spot; (**c**) multiplex original laser spot; (**d**) multiplex received laser spot.

**Figure 3 sensors-23-08737-f003:**
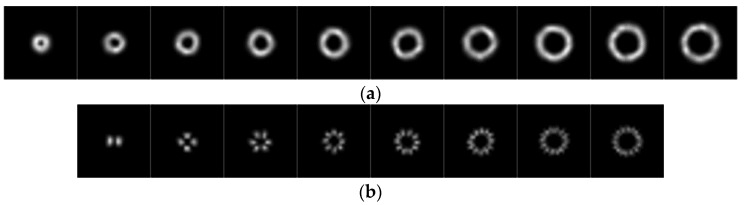
Intensity images of received LG beam. (**a**) $\left|l_{0}\right|=1,2,\ldots 10$ (single OAM). (**b**) $\left|l_{0}\right|=1,2,\ldots 8$ (multiplexed OAM).

**Figure 4 sensors-23-08737-f004:**
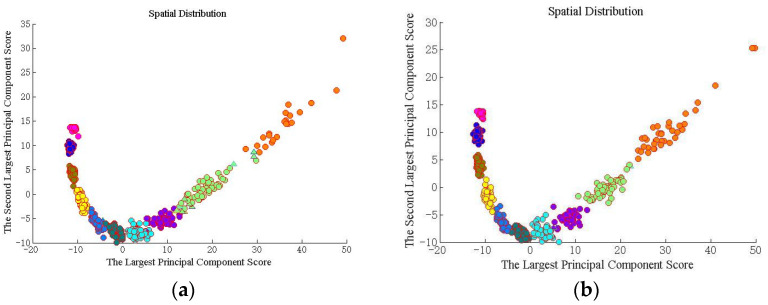
Spatial distribution for a single topological charges dataset. (**a**) $C_{n}^{2}=1\times 10^{-14}\;m^{-2/3}$. (**b**) $C_{n}^{2}=1\times 10^{-15}\;m^{-2/3}$.

**Figure 5 sensors-23-08737-f005:**
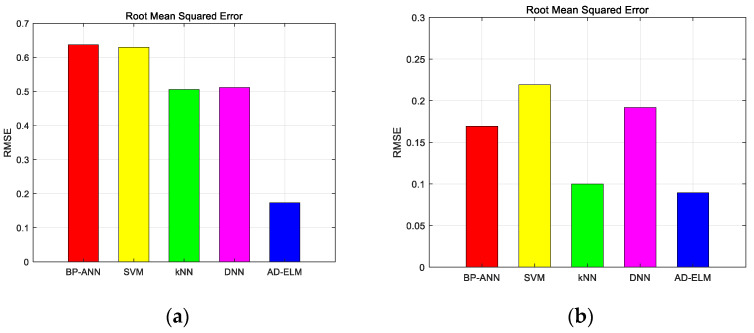
RMSE of the five methods for single topological charges dataset. (**a**) $C_{n}^{2}=1\times 10^{-14}\;m^{-2/3}$; (**b**) $C_{n}^{2}=1\times 10^{-15}\;m^{-2/3}$.

**Figure 6 sensors-23-08737-f006:**
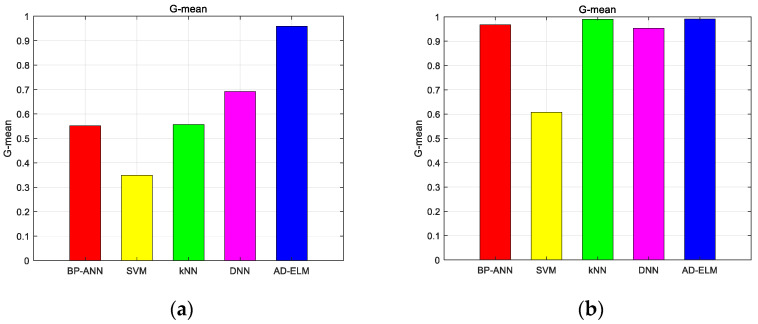
G-mean of the five methods for single topological charges dataset. (**a**) $C_{n}^{2}=1\times 10^{-14}\;m^{-2/3}$; (**b**) $C_{n}^{2}=1\times 10^{-15}\;m^{-2/3}$.

**Figure 7 sensors-23-08737-f007:**
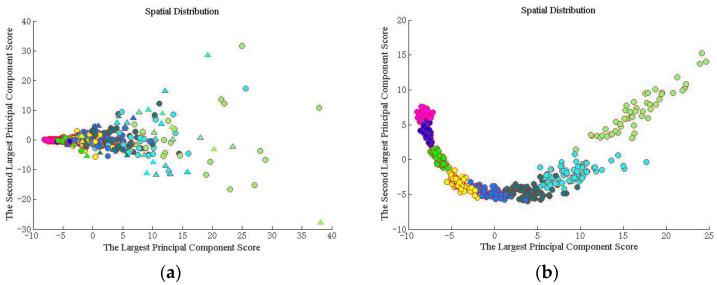
Spatial distribution for multiplexed topological charges dataset. (**a**) $C_{n}^{2}=1\times 10^{-14}\;m^{-2/3}$; (**b**) $C_{n}^{2}=1\times 10^{-15}\;m^{-2/3}$.

**Figure 8 sensors-23-08737-f008:**
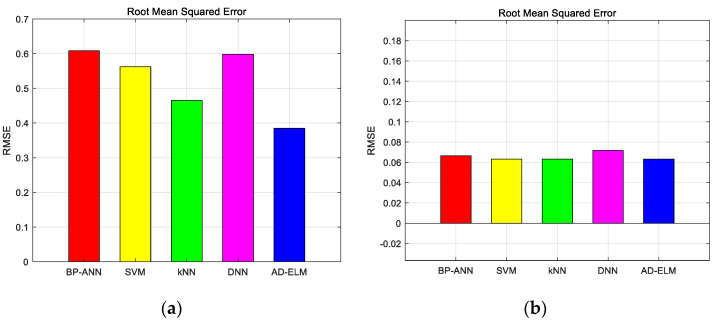
RMSE of the five methods for multiplexed topological charges dataset. (**a**) $C_{n}^{2}=1\times 10^{-14}\;m^{-2/3}$; (**b**) $C_{n}^{2}=1\times 10^{-15}\;m^{-2/3}$.

**Figure 9 sensors-23-08737-f009:**
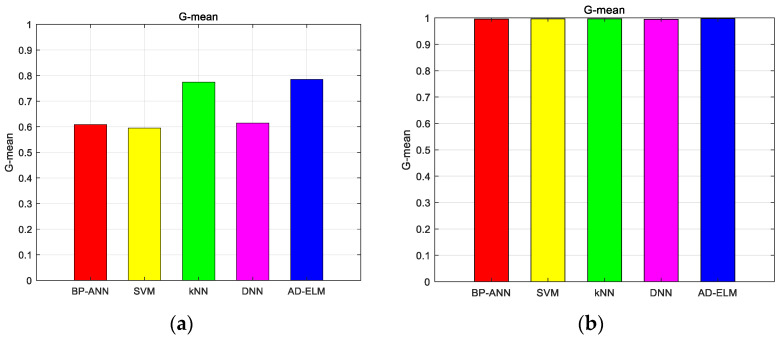
G-mean of the five methods for multiplexed topological charges dataset. (**a**) $C_{n}^{2}=1\times 10^{-14}\;m^{-2/3}$; (**b**) $C_{n}^{2}=1\times 10^{-15}\;m^{-2/3}$.

**Table 1 sensors-23-08737-t001:** Confusion matrix.

Actual Result	Predicted Result	
	Positive	Negative
Positive	TP (*True Positive*)	FN (*False Negative*)
Negative	FP (*False Positive*)	TN (*True Negative*)

## Data Availability

The data presented in this study are available in Appendix A.

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
