# Peer review of "An Efficient Recognition Method for Orbital Angular Momentum via Adaptive Deep ELM"

_sensors, 2023, doi:10.3390/s23218737_

Round 1

Reviewer 1 Report

The manuscript "An Efficient Recognition Method for Orbital Angular Momentum via Adaptive Deep ELM" by Haiyang Yu et al. provides an interesting exploration of using machine learning for discerning Orbital Angular Momentum (OAM) states after atmospheric turbulence. This work has notable implications for improving OAM-based communication. However, the following issues and areas of clarification should be addressed to improve the manuscript:

1. The manuscript can be challenging for non-specialist readers to comprehend. For example, it's unclear how the "largest principal component scores" are calculated. The meaning of different color blocks in Figures 4 and 7 could also be more explicitly explained. Clarification of these points could help to broaden the manuscript's accessibility.

2. The selection of structure constant values (10^-14 and 10^-15) as characteristic parameters for air turbulence lacks sufficient justification. The relationship between these values and real-world atmospheric turbulence is not established. It would be beneficial to explain why these specific values were chosen and whether other values could be suitable.

3. The term "experiment results" may be misleading, as it appears the results are from simulations rather than actual physical experiments. Consider using a more precise term, such as "simulation results," to avoid potential confusion. 

4. The distribution of color blocks in Figure 7(a) differs significantly from similar figures. Once the origins and significance of these data are clarified, further explanation of this notable difference would be helpful.

Author Response

Responses to reviewers’ comments

Dear Editor and anonymous reviewers,

In this document, we describe how we have taken the reviewers’ comments into consideration and revised the manuscript. We include the reviewers’ comments and questions in small black italic font, our responses are in large, blue bold-face font, and excerpts from our revised manuscript in underlined blue italic font.

We have addressed each of the individual comments from each reviewer and hope that this revision clarifies any remaining concerns.

We would like to thank the reviewers and the editor for their suggestions and feedback on improving the presentation of our manuscript.

Sincerely,

Haiyang Yu

-------------------------------------------------------------------------------------------------------

Paper ID: sensors-2602590

Paper Title: An Efficient Recognition Method for Orbital Angular Momentum via Adaptive Deep ELM

Comments from the Reviewers

  1. The manuscript can be challenging for non-specialist readers to comprehend. For example, it's unclear how the "largest principal component scores" are calculated. The meaning of different color blocks in Figures 4 and 7 could also be more explicitly explained. Clarification of these points could help to broaden the manuscript's accessibility.

Thanks very much for your constructive comment. We have explained the notions from Line 283 to 292, which is described with highlighted. We first calculated the principal components, and then sorted them from the largest to the smallest. After calculated the eigenvectors for each sample separately, We use blue bars to represent the proposed method, and the red ones, the yellow ones, the green ones, and the pink ones represent BPP-ANN, SVM, kNN, DNN, respectively.

  1. The selection of structure constant values (10^-14 and 10^-15) as characteristic parameters for air turbulence lacks sufficient justification. The relationship between these values and real-world atmospheric turbulence is not established. It would be beneficial to explain why these specific values were chosen and whether other values could be suitable.

Thanks very much for your detailed comment. We have added reference [31] and quoted it. The paper described the relationship between these values and real-world atmospheric turbulence. Davis proposed a method to divide the strength of turbulence according to[31]. When the , the atmospheric turbulence is weak turbulence. As  increases, the atmospheric turbulence gradually becomes stronger.(from Line 240 to 2444)

  1. 31. Davis J. Consideration of atmospheric turbulence in laser systems design. Applied Optics s, 1966, 5(1), 139-147.

  1. The term "experiment results" may be misleading, as it appears the results are from simulations rather than actual physical experiments. Consider using a more precise term, such as "simulation results," to avoid potential confusion.

Thanks very much for your detailed comment. We have changed the experiment results to simulation results.

  1. The distribution of color blocks in Figure 7(a) differs significantly from similar figures. Once the origins and significance of these data are clarified, further explanation of this notable difference would be helpful.

Thanks very much for your detailed comment. The different color spots with various color circles show the different datasets.

Reviewer 2 Report

None

Author Response

Responses to reviewers’ comments

Dear Editor and anonymous reviewers,

In this document, we describe how we have taken the reviewers’ comments into consideration and revised the manuscript. We include the reviewers’ comments and questions in small black italic font, our responses are in large, blue bold-face font, and excerpts from our revised manuscript in underlined blue italic font.

We have addressed each of the individual comments from each reviewer and hope that this revision clarifies any remaining concerns.

We would like to thank the reviewers and the editor for their suggestions and feedback on improving the presentation of our manuscript.

Sincerely,

Haiyang Yu

-------------------------------------------------------------------------------------------------------

Paper ID: sensors-2602590

Paper Title: An Efficient Recognition Method for Orbital Angular Momentum via Adaptive Deep ELM

Comments from the Reviewers

  1. In lines 10-11, the expressions are unclear.

Thanks very much for your constructive comment. We have added the abbreviations explanation and the expressions are normalized.

  1. The authors mainly demonstrate the recognition of the laser spots. However, how to define OAMs using the laser spots? How to implement the OAM mode label using the laser spots?

Thanks very much for your constructive comment. We have explained the OAM mode.  is the topological charge. Otherwise  is known as spatial modes. The number of  is OAM mode label.(at Line 144). The implement the OAM mode label use numerical simulation. We have changed the experiment results to simulation results.

  1. In Fig.1, how do optical signals convert into electronic signal, to complete neural network computing?

Thanks very much for your constructive comment. We implemented numerical simulation and received light spots through a phase screen. .(at Line 135).

  1. How does the distance of space influence the NSME under different neural network methods?

Thanks very much for your constructive comment. We have not the distance too much. We define the 2000m condition. We simulate the 2000m transmission distance. .(at Line 237). We hope to first verify the feasibility of ELM application to OAM recognition.

Reviewer 3 Report

The authors presented a very well detailed work, with a clear digression covering all the aspects of their research.
The work appears to represent a mere improvement of already stated methods, but the overall presentation, the structuring of the elaboration framework, and the data management combined with the peculiar structuring of the DL setup deserve pubblication, even because the manuscript is well organized.
Here there are only minor suggestions:
1) Please, also define "extreme learning machine" (ELM) in the abstract
2) Some formulas requires just easy and small fixings; please, check them carefully (for instance, maybe l must be replaced with l0 into the exp of the 2nd line of (7))
3) The Eniglish is clear and good; it just needs some small fixings all around (for instance, "When the turbulent changed weak", line 377; maybe a recast is needed)
4) This is maybe one important point to address: every DL/ML method requires lots of data, which is the true key factor to achieve reliability in these protocols; unfortunately, abundant dataset mean also large amount of time to recover them and more properly also large time to process the same data in the DL evaluation stage.
The authors must provide clear and well detailed description on the time & the resources needed to assemble the training dataset, together with a similar but even larger, clearer and more detialed description of the training stage; this aspect plays a crucial role in this research and cannot be negleted for no reason.

There are no hidden comments for the Edtor(s).

Author Response

Responses to reviewers’ comments

Dear Editor and anonymous reviewers,

In this document, we describe how we have taken the reviewers’ comments into consideration and revised the manuscript. We include the reviewers’ comments and questions in small black italic font, our responses are in large, blue bold-face font, and excerpts from our revised manuscript in underlined blue italic font.

We have addressed each of the individual comments from each reviewer and hope that this revision clarifies any remaining concerns.

We would like to thank the reviewers and the editor for their suggestions and feedback on improving the presentation of our manuscript.

Sincerely,

Haiyang Yu

-------------------------------------------------------------------------------------------------------

Paper ID: sensors-2602590

Paper Title: An Efficient Recognition Method for Orbital Angular Momentum via Adaptive Deep ELM

Comments from the Reviewers

Here there are only minor suggestions:
1) Please, also define "extreme learning machine" (ELM) in the abstract

Thanks very much for your constructive comment. We have added the abbreviations explanation and the expressions are normalized.

2) Some formulas requires just easy and small fixings; please, check them carefully (for instance, maybe l must be replaced with l0 into the exp of the 2nd line of (7))

Thanks very much for your constructive comment. We have corrected some formulas.

 ,            (7)

3) The Eniglish is clear and good; it just needs some small fixings all around (for instance, "When the turbulent changed weak", line 377; maybe a recast is needed)

Thanks very much for your constructive comment. We have improved some expression. When the turbulence changed weak.(at Line 319).

4) This is maybe one important point to address: every DL/ML method requires lots of data, which is the true key factor to achieve reliability in these protocols; unfortunately, abundant dataset mean also large amount of time to recover them and more properly also large time to process the same data in the DL evaluation stage.
The authors must provide clear and well detailed description on the time & the resources needed to assemble the training dataset, together with a similar but even larger, clearer and more detialed description of the training stage; this aspect plays a crucial role in this research and cannot be negleted for no reason.

Thanks very much for your constructive comment. We further describe the data set.Davis proposed a method to divide the strength of turbulence according to [31]. When the , the atmospheric turbulence is weak turbulence. As  increases, the atmospheric turbulence gradually becomes stronger. In order to evaluate the effectiveness of the proposed method, we generate different datasets which  are  and ,respectively.  After image processing, the received laser spot images retain the key characteristic information, and the spot size of each attribute is . We simulated 2500 OAM modal samples. Here the 2000 training samples and 500 test samples are randomly collected as the input of the model. (at Line from 243 to 251).

  1. 31. Davis J. Consideration of atmospheric turbulence in laser systems design. Applied Optics s, 1966, 5(1), 139-147.

Reviewer 4 Report

In this manuscript, the authors introduce a novel deep extreme learning machine (ELM) for recognizing orbital angular momentum (OAM). It outperforms other methods like KNN and SVM on simulated datasets. The paper is well-organized and the results are of interest. I suggest its publication after minor revisions.  My concerns are listed as follows.

1) The authors should explain the comparison methods (SVM, KNN and others) more thoroughly to ensure the better performance of the proposed method isn't solely due to a higher model capacity.

2) In Fig4, legends are needed to indicate the corresponding classes.

3) Can one use this proposed method for unmixing or demodulating OAM carriers in the same channel?

4) Reference should be given for the parameter selection of C_n = 10^-14 and 10^-15.

Author Response

Responses to reviewers’ comments

Dear Editor and anonymous reviewers,

In this document, we describe how we have taken the reviewers’ comments into consideration and revised the manuscript. We include the reviewers’ comments and questions in small black italic font, our responses are in large, blue bold-face font, and excerpts from our revised manuscript in underlined blue italic font.

We have addressed each of the individual comments from each reviewer and hope that this revision clarifies any remaining concerns.

We would like to thank the reviewers and the editor for their suggestions and feedback on improving the presentation of our manuscript.

Sincerely,

Haiyang Yu

-------------------------------------------------------------------------------------------------------

Paper ID: sensors-2602590

Paper Title: An Efficient Recognition Method for Orbital Angular Momentum via Adaptive Deep ELM

Comments from the Reviewers

1) The authors should explain the comparison methods (SVM, KNN and others) more thoroughly to ensure the better performance of the proposed method isn't solely due to a higher model capacity.

Thanks very much for your detailed comment. We further explain the advantages of the proposed method compared with the comparative method. The analytical solutions and the absence of manual parameters are the key elements for capability improvement. When the turbulence turns strong, the proposed AD-ELM also get the optimal model to match the changing environment. At the same time, a better analytical expression is given by the AD-ELM, which represent a higher model capacity without manual trial and error operation, which have significant application value for long-term online recognition. The optimal parameters help our method more suitable for the regular OAM mode sets. .(at Line from 332 to 337).

2) In Fig4, legends are needed to indicate the corresponding classes.

Thanks very much for your detailed comment. The different color spots with various color circles show the different datasets.

3) Can one use this proposed method for unmixing or demodulating OAM carriers in the same channel?

Thanks very much for your detailed comment. We have changed the experiment results to simulation results. We hope to first verify the feasibility of ELM application to OAM recognition. So we looked at the results of numerical simulations.We simulate the 2000m transmission distance. .(at Line 237).

4) Reference should be given for the parameter selection of C_n = 10^-14 and 10^-15.

Thanks very much for your detailed comment. We have added reference [31] and quoted it. The paper described the relationship between these values and real-world atmospheric turbulence. Davis proposed a method to divide the strength of turbulence according to[31]. When the , the atmospheric turbulence is weak turbulence. As  increases, the atmospheric turbulence gradually becomes stronger.(from Line 240 to 2444)

  1. 31. Davis J. Consideration of atmospheric turbulence in laser systems design. Applied Optics s, 1966, 5(1), 139-147.

Round 2

Reviewer 2 Report

Authors have well reponsed the issues proposed by the reviewers. I suggest that this manuscript will be published